# Novel HSP90-PI3K Dual Inhibitor Suppresses Melanoma Cell Proliferation by Interfering with HSP90-EGFR Interaction and Downstream Signaling Pathways

**DOI:** 10.3390/ijms21051845

**Published:** 2020-03-07

**Authors:** Qian Zhao, Hong-Ping Zhu, Xin Xie, Qing Mao, Yan-Qing Liu, Xiang-Hong He, Cheng Peng, Qing-Lin Jiang, Wei Huang

**Affiliations:** 1State Key Laboratory of Southwestern Chinese Medicine Resources, School of Pharmacy, Chengdu University of Traditional Chinese Medicine, Chengdu 611137, China; zhaoq584@sina.cn (Q.Z.); xiexin@cdutcm.edu.cn (X.X.); maoqing0808@126.com (Q.M.); lyq82893@163.com (Y.-Q.L.); hexianghong@stu.cdutcm.edu.cn (X.-H.H.); pengcheng@cdutcm.edu.cn (C.P.); 2Antibiotics Research and Re-evaluation Key Laboratory of Sichuan Province, Sichuan Industrial Institute of Antibiotics, Chengdu University, Chengdu 610052, China; zhuhp517@hotmail.com; 3Sichuan Province College Key Laboratory of Structure-Specific Small Molecule Drugs, School of Pharmacy, Chengdu Medical College, Chengdu 610500, China

**Keywords:** Hsp90, PI3K, apoptosis, pyroptosis, melanoma

## Abstract

Melanoma is the deadliest form of skin cancer, and its incidence has continuously increased over the past 20 years. Therefore, the discovery of a novel targeted therapeutic strategy for melanoma is urgently needed. In our study, MTT-based cell proliferation assay, cell cycle, and apoptosis assays through flow cytometry, protein immunoblotting, protein immunoprecipitation, designing of melanoma xenograft models, and immunohistochemical/immunofluorescent assays were carried out to determine the detailed molecular mechanisms of a novel HSP90-PI3K dual inhibitor. Our compound, named DHP1808, was found to suppress A375 cell proliferation through apoptosis induction by activating the Fas/FasL signaling pathway; it also induced cell-cycle arrest and inhibited the cell migration and invasion of A375 cells by interfering with Hsp90-EGFR interactions and downstream signaling pathways. Our results indicate that DHP1808 could be a promising lead compound for the Hsp90/PI3K dual inhibitor.

## 1. Introduction

Melanoma is the deadliest form of skin cancer worldwide, with more than 96,480 new cases and 7230 cancer-related deaths in the United States in 2019 [1]. The incidence rates for melanoma have been on a continuous rise over the past 20 years. Surgical resection is the main therapeutic approach for patients with early melanoma. Given the high malignancy degree of patients with melanoma in advanced stages, the effect of standardized treatment is poor, and the accompanying diagnosis and individualized targeted therapy have become important strategies [1,2,3,4,5,6,7,8]. Several kinase inhibitors, including BRAF inhibitor vemurafenib and dabrafenib, MEK1/2 inhibitor trametinib, *c*-kit inhibitor imatinib and nilotinib, and other inhibitors targeted to the PI3K/Akt/mTOR signaling pathways, have been approved for the treatment of metastatic melanoma [9,10,11,12,13].

Heat shock protein 90 (Hsp90) is a kind of molecular chaperone protein that functions in client protein folding processes to maintain correct protein conformations [14,15,16,17,18]. Inhibiting the interactions between Hsp90 and chaperone proteins usually causes ubiquitination and subsequent proteasome degradation of client proteins. The discovery of Hsp90 inhibitors had been an attractive strategy for cancer target therapy. Several structurally diverse Hsp90 inhibitors, including geldanamycin, SNX-2112, and AUY-922, had entered clinical trial stages [19,20,21]. The phosphoinositide 3-kinase (PI3K) signaling pathway is one of the most commonly dysregulated pathways in advanced melanoma. The PI3K pathway is often activated by the silence or genetic loss of phosphatase and tensin homolog and/or the overexpression of protein kinase B [22,23]. The PI3K pathway plays a vital role in carcinogenesis, drug resistance, and programmed cell death of melanoma cells [24,25,26,27,28,29]. Therefore, this pathway is a promising target for melanoma therapy. In the analysis of the mRNA expression profiles of Hsp90 and PI3K in patients with melanoma, we found that their expression levels were positively correlated in melanoma tissues. Thus, the combination of Hsp90 and PI3K inhibition might synergistically suppress the proliferation, metastasis, and survival signals of melanoma.

In the current study, we investigated the anti-proliferation, apoptosis, pytoptosis-inducing potencies, and potential molecular mechanism of a novel Hsp90/PI3K inhibitor DHP1808 on A375 melanoma cells in vitro and in vivo. The results demonstrated that the Hsp90/PI3K dual inhibitor efficiently induced melanoma cell apoptotic death by interfering with the Hsp90-EGFR interaction and suppressed the downstream MAPK signaling pathways. Furthermore, DHP1808 induced less pyroptosis, compared with the combination of Hsp90 and PI3K inhibitors in tumor and intestinal tissues, suggesting that these dual inhibitors can serve as safe drugs and are worthy of further development.

## 2. Results

### 2.1. HSP90 and PI3Kα Are Significantly Overexpressed in Melanoma with Positive Correlationship

We analyzed the correlation between HSP90AA1 TPM and PIK3CA TPM, as shown in Appendix A. The results show a strong correlation between the changes in HSP90AA1 TPM and PIK3CA TPM, with high reliability of correlation. Appendix A shows the expression of HSP90 AA1 and PIK3CA in normal human skin. In addition, we also analyzed sequences from patients with skin cutaneous melanoma in the cBioPortal database [30,31,32,33,34]. We found that 10% and 19% patients had mutations or dysregulated expression on HSP90AA1 and PIK3CA, respectively (Appendix A). The HSP90AA1 mutation was one of the main factors behind amplified expression of HSP90AA1 mRNA, whereas dysregulated PIK3CA mRNA levels were not significantly correlated to its gene mutation.

### 2.2. Novel HSP90/PI3Ka Dual Inhibitor DHP1808 Suppresses A375 Cell Proliferation and Induces Cell Death

The dual inhibitor DHP1808 with a novel structure is shown in Figure 1A. Its half maximal inhibitory concentrations against Hsp90 and PI3Kα were at sub-micromolar levels, as reported in our recent reports [21,35,36,37,38,39]. No obvious off-target kinase inhibition was observed (Appendix A). Its cytotoxicity was initially evaluated in a panel of human melanoma cell lines, which were incubated with different doses of DHP1808 for 24 h. Cell viability was measured by MTT assay. As shown in Figure 1B, the cytotoxic activity of DHP1808 is remarkably higher in A375 than in others, with IC_50_ of approximately 2.29 μg/mL after 24 h of treatment. We further treated A375 cells with varying concentrations of DHP1808 for 5 days, allowing colony formation to explore the detailed mechanism of DHP1808. Figure 1C (S2) shows that in the DHP1808-incubated A375 cells, the number of colonies significantly decreased, compared to that of the control group. When the concentration of DHP1808 reached 0.3 μg/mL, the number of colonies was completely suppressed, indicating that DHP1808 caused a dose-dependent inhibition of colonies. These results suggested DHP1808 may inhibit cell proliferation and induce cell death.

### 2.3. DHP1808 Induces A375 Cell Apoptosis by Activating the Fas/FasL Signaling Pathway

Hoechst 33,258 staining was used to investigate morphological changes in DHP1808-treated A375 cells to assess cell death and apoptosis. Microscopy revealed that the apoptotic nuclei condensed and fragmented after 24 h of therapy (Appendix A). Subsequent flow cytometry experiments with Annexin V/PI dual staining was performed to examine the activation of apoptosis and investigate the possibility of cell death induced by DHP1808 (Figure 2A and Appendix A). The apoptotic cells evidently increased after DHP1808 was incubated for 24 h, and the percentage of Annexin V-positive apoptotic cells treated with 20 μg/mL (42.6 ± 6.30%) of DHP1808 (35.7 ± 4.50%) was significantly higher than that with 40 μg/mL (21.7 ± 4.26%) treatment or in the absence of the compound (2.7 ± 0.2%, *p* < 0.05). As such, DHP1808 induced A375 and SK-Mel-28 cell apoptosis in an increasing dose-dependent manner, compared with the control group. However, apoptosis did not vary when the concentration of DHP1808 varied from 2.5 to 10 μg/mL.

We studied anti-apoptotic and pro-apoptotic protein expression to further explore the mechanism by which DHP1808 induced cell apoptosis in A375 and SK-Mel-28 cells. Western blot analysis results showed (Figure 2B and Appendix A) that treating A375 cells with DHP1808 (20 and 40 μg/mL) remarkably upregulated the cleaved caspase-3, caspase-8, caspase-9, and PARP expression; the expression levels of Fas and FasL were upregulated. However, the levels of cytochrome C, FADD, Bcl-2, Bax, or Bad were not altered. In a typical procedure, these findings indicate that DHP1808 induces apoptosis by activating the Fas/FasL signaling pathways in A375 cells.

### 2.4. DHP1808 Induces Cell Cycle Arrest and Inhibits A375 Cell Migration and Invasion

Given that our previous data indicated that DHP1808 exhibited a potent effect on melanoma cell proliferation and survival, we studied the effect of DHP1808 on cell-cycle progression. Flow cytometry analyses confirmed that DHP1808 also induced cell-cycle arrest in A375 cells. Cell counts in the G2 phase were remarkably increased after incubation with DHP1808 for 24 h, whereas cell counts in the G1 phase decreased (Figure 3A). Low concentrations of the drug were sufficient to arrest cells in the G2 phase. These results were demonstrated by the overexpression of p21 and p27 and the reduction of CyclinB1, CDK2, and CDK6 proteins compared with those in the control group (Figure 3B).

Transwell and agarose wound healing assays were performed to investigate whether DHP1808 was involved in inhibiting the invasion and migration of melanoma cells. As shown in Appendix A, cell migration in A375 cells significantly decreased in a dose-dependent manner on treatment with DHP1808. We then performed a wound-healing assay to further illustrate the effects of DHP1808 on cell motility (Appendix A). The wound areas of A375 and SK-Mel-28 cells had minimal changes after 15 μg/mL of DHP1808 incubation, compared with those of the control plates, and increasing the concentration of DHP1808 did not change the result. We also investigated the expression levels of tumor invasion- and migration-associated proteins at different concentrations of DHP1808 (20 and 40 μg/mL). The levels of β-catenin and E-Cad were not changed after DHP1808 was applied. In contrast, the expression of N-Cad, Vim, MMP-2, MMP-9, and ZEB1 was significantly decreased in the tumors (Figure 3C). Therefore, DHP1808 can alter the expression level of proteins associated with the migration and invasion of A375 cells.

### 2.5. DHP1808 Inhibits the Interaction between HSP90 and EGFR

Drug affinity responsive target stability strategy has demonstrated that HSP90 is the direct target of DHP1808, but its molecular mechanism in melanoma cells remains unclear. We performed co-immunoprecipitation among HSP90 and EGFR to unravel the interaction between HSP90 and DHP1808. After A375 cells were treated with 0, 20, and 40 ug/mL of DHP1808 for 24 h, HSP90 was found in the HSP90 complex captured by the HSP90 antibody in the same amount as that of the input. However, no EGFR protein was detected in the HSP90 complex after DHP1808 treatment (Figure 4A). Reverse co-precipitation was also performed showing that EGFR hardly co-precipitated with HSP90. However, the total amount of HSP90 was obviously more than that in EGFR immunoprecipitated by the HSP90 antibody (Figure 4B). As such, DHP1808 can significantly block the combination of HSP90 and EGFR in A375 cells.

### 2.6. DHP1808 Induces the PI3K/Akt Signaling Pathway and Regulate the Signaling Pathways Related to HSP90 Client Proteins

The expression levels of the HSP90 family proteins and the PI3K downstream-regulated proteins were determined by western blot analysis of A375 melanoma cell extracts to explain the dual inhibitory effects of DHP1808. We first determined the effect of DHP1808 on the HSP90-CDC37 chaperone system and examined the expression of HSP90 client proteins in DHP1808-treated A375 cells (Figure 5A). DHP1808 treatment reduced the protein levels of cRaf, bRaf, CDK4, EGFR, and the phosphorylated Met, cRaf, and bRaf. HSP90 inhibited might have degraded kinase clients, thereby improving therapeutic efficacy against tumors. The HSP90-CDC37 chaperone system is essential for the integrity and viability of the signaling pathways involved in cell-cycle control and organism development by mediating the maturation of protein kinase clients and supporting kinase functional activity.

The MAPK and PI3K-Akt signaling pathways are important signaling pathways related to HSP90 client proteins that stimulate cell proliferation, oncogene transcription, cell invasion, and migration. Our previous data proved that DHP1808 inhibited HSP90, causing the degradation of kinase clients; thus, we investigated the expression of proteins associated with the MAPK signaling pathway (Figure 5B). Western blot analysis showed a decrease in c-myc, p90RSK, and phosphorylated p90RSK expression, further indicating that DHP1808 can inhibit cell proliferation and induce cell apoptosis by suppressing the MAPK/ERK signaling pathway. However, DHP1808 did not alter the ERK1/2 and β-catenin expression and the phosphorylation of ERK1/2 in A375 cells. DHP1808 reduced the levels of total Akt expression as a result of the PI3K pathway inhibition, causing a significant increase in the downstream protein pro-apoptosis factor Bim (Figure 5C). Nevertheless, DHP1808 did not substantially affect the phosphorylation of Akt473 and Akt308 (Figure 5C). DHP1808 decreased the expression of MDM2, thereby increasing p53 (Figure 5C). The expression of EGFR and that of the DHP1808-treated or non-treated group did not vary (Figure 5D). Thus, DHP1808 can efficiently regulate HSP90 client proteins and the MAPK and PI3K-Akt signaling pathways.

### 2.7. DHP1808 Resulted in Less Pyroptosis than the Combination of HSP90 and PI3K Inhibitors

Several reports suggest that Hsp90 and HDACs are involved in the pyroptosis of tumor cells and tumor-associated inflammation [40,41,42,43,44,45]. Although the pro-tumorigenic and anti-tumorigenic effects of pyroptosis have been reported in various types of cancers, gasdermin E (GSDME)-mediated pyroptosis might result in normal tissue toxicity after chemotherapy [46,47,48,49,50]. The knockdown of GSDME expression is likely mitigated by the chemotherapeutic adverse effects in normal tissues [51,52,53]. LDH release, western blotting, and immunofluorescent assays were performed to study whether the combination of the Hsp90 and PI3K inhibitors mediated more pyroptotic cell death than that of the Hsp90/PI3K dual inhibitor.

As shown in Figure 6A, caspase-1 and GSDME were not significantly activated after DHP1808, Hsp90 inhibitor AUY-922, and pan-PI3K inhibitor PI-103 incubation. However, the combination of AUY-922 and PI-103 evidently activated the cleavage activation of caspase-1 and GSDME, suggesting the initiation of pyroptosis. The results of LDH release assays were consistent with those of protein immunoblotting assays (Figure 6B). In the GSDME immunofluorescent assays, pre-treatment of melanoma cells with caspase-1 inhibitor VX-765 or pan-caspase inhibitor Z-VAD-FMK suppressed the GSDME activation induced by the combined Hsp90 and PI3K inhibitors (Figure 6C). Altogether, our results suggest that DHP1808 induced less pyroptotic cell death than that of the combined Hsp90 and PI3K inhibitors.

### 2.8. DHP1808 Inhibited Tumor Growth In Vivo

An A375 subcutaneous xenograft model in female nude mice was established to investigate the anti-melanoma effect of DHP1808 in vivo. DHP1808 administration suppressed the growth of mouse xenografts (*p* < 0.01) (Figure 7A). The average weight of the melanomas from the treatment group was significantly lower than that of the control group (Figure 7B). The inhibition potencies of the xenograft tumor volume and weight in the DHP1808 group were comparable to those of the AUY-922 plus PI-103 group. However, the body weight of mice in AUY-922 plus PI-103 group continuously declined from day 9 to day 21, whereas the changes in DHP1808, AUY-922, and PI-103 groups were not significant (Figure 7C). As shown in Figure 7D, the immunofluorescent assay of cleaved GSDME on the intestinal tissues in each group suggested that the activation of GSDME in AUY-922 plus PI-103 group was stronger than that of the other groups. We used immunofluorescence and IHC to analyze tumor tissues to further verify whether the tumor growth suppression regulated by DHP1808 was related to the inhibition of cell proliferation and the dual inhibition of Hsp90/PI3K. As shown in Figure 7E, the number of apoptotic positive cells determined by TUNEL staining the DHP1808-treated tumors was significant. In addition, IHC analysis proved that the expression of the PI3K downstream protein Akt and the Hsp90 client protein EGFR decreased, and the proliferative marker Ki-67 was inhibited. Treatment with 15 mg/kg of DHP1808 can evidently inhibit the phosphorylation level of Akt protein; however, the inhibitory effect of PI-103 was weakened by the combination of AUY-922 (*p* < 0.01). The animal experiments showed that DHP1808 exhibited excellent inhibitory effects on HSP90 and PI3K in vivo. Taking such results together, the results of the in vivo experiments were in accordance with those of the in vitro experiments.

## 3. Discussion

The combination of PI3K inhibitors and other kinase inhibitors has been frequently reported to display synergistic effects against various tumors, including melanoma [38,54,55,56,57]. In this work, we evaluated the therapeutic potency and molecular mechanism of a novel Hsp90/PI3K inhibitor, DHP1808, in BRAF V600E-mutated A375 melanoma cells. After the mRNA expression profiles of Hsp90 and PI3K in melanoma tissues and normal skin tissues were analyzed, we found that the expression of Hsp90 and PI3K was strongly correlated in melanoma tissues compared to that in normal skin tissues. Thus, the dual inhibitors targeting Hsp90 and PI3K probably exerted synergistic effects against melanoma cell proliferation. On the basis of our previous research on the design and synthesis of novel Hsp90/PI3K dual inhibitors, DHP1808 was synthesized and selected for subsequent mechanistic studies. Although the results of Hoechst 33,258 staining and Annexin V/PI dual staining indicated that DHP1808 incubation induced remarkable apoptosis in A375 cells, mitochondrial apoptosis markers, such as Bax, Bad, and cytochrome C, were not evidently changed. The proteins related to the death receptor pathway, including Fas and FasL, were significantly activated after DHP1808 treatment, suggesting that extrinsic apoptosis might be involved in the DHP1808-mediated apoptosis.

As a molecular chaperone protein, Hsp90 is essential to its client proteins because it maintains their stability and prevents their degradation from proteasome [58,59,60]. In the protein immunoparticipation assays, the addition of DHP1808 interfered with the interactions between Hsp90 and EGFR, resulting in the degradation of EGFR. The EGFR downstream signaling MAPK pathways, including the ERK, JNK, and MNK pathways, were inhibited by DHP1808 in a dose-dependent manner. PI3K inhibition by DHP1808 resulted in the declined phosphorylation levels of Akt and mTOR. The inhibition of the PI3K-Akt-mTOR pathway also contributed to the negative regulation of the MAPK pathways.

Pyroptosis is a type of programmed cell death subroutines and is characterized by the release of IL-1β and IL-18 to recruit immune cells and initiate defensive inflammatory response [61,62,63]. The roles of pyroptosis in cancer progression and drug resistance are diverse. Lage et al. reported that the loss of GSDME in melanoma cells resulted in resistance to etoposide incubation [64,65]. Expression was decreased in etoposide-resistant melanoma cells and was negatively associated with cell resistance to etoposide-induced cell death. However, knockout of GSDME protected mice from various organ toxicities and body weight loss induced by chemotherapy drugs [66,67,68]. In the current study, DHP1808 induced less pyroptotic cell death than that of the combined Hsp90 and PI3K inhibitors in vitro and in vivo. In consideration of the fact that DHP1808 a displayed similar antitumor effect to the combination of Hsp90 and PI3K inhibitors in the xenograft models, we speculate that this novel Hsp90/PI3K dual inhibitor is safe to use and should be further developed.

In conclusion, we reported the anti-proliferation effect and molecular mechanism of a novel Hsp90/PI3Kα dual inhibitor DHP1808 in A375 melanoma cells. DHP1808 remarkably suppressed A375 cell proliferation, migration, and invasion; it also induced cell-cycle arrest and apoptotic cell death. DHP1808 inhibited the MAPK signaling pathway activation by synergistically interfering with Hsp90-EGFR interaction and suppressing the PI3K-Akt pathway. Furthermore, DHP1808 induced less pyroptosis in the tumor and normal tissues than the combination of Hsp90 and PI3K inhibitors, thereby indicating drug safety. Thus, the novel Hsp90/PI3Kα dual inhibitor might be a potential drug candidate for the targeted therapy of melanoma.

## 4. Materials and Methods

### 4.1. Reagents and Antibodies

The preparation of DHP1808 was as per our previous reports [38]. VX-765 and Z-VAD-FMK were obtained from Selleckchem Co. Ltd. (Shanghai, China). The antibodies recognizing FADD (14906-1-AP), Bcl-2 (12789-1-AP), Bax (50599-1-AP), Cytochrome C (10993-1-AP), β-Catenin (51067-2-AP), CDK2 (10122-1-AP), CDK4 (11026-1-AP), CDK6 (14052-1-AP), CyclinB1 (55004-1-AP), p21 (10355-1-AP), E-Cadherin (20874-1-AP), MMP2 (10373-1-AP), MMP9 (10375-1-AP), ZEB1 (21544-1-AP), GAPDH (60004-1-Ig) and β-actin (60008-1-Ig), MNK1 (10136-1-AP), MDM2 (19058-1-AP), and p53 (10442-1-AP) were purchased from Proteintech (Wuhan, China). The antibody recognizing Fas (Ab82419), FasL (Ab68338), Caspase-3 (Ab13847), PARP (Ab32138), N-Cadherin (Ab76011), Met (Ab51067), Phos-Met (Ab68141), ERK1/2 (Ab17942), Phos-ERK1/2 (Ab76299), JNK (Ab179461), Phos-JNK (Ab124956), phos-MNK1 (ab109102), Caspase-1 (ab179515), and GSDMD (ab215203) were purchased from Abcam (Cambridge, MA, USA). The antibody recognizing Bad (9239), Bim (2933), Caspase-8 (9746), Caspase-9 (9508), p27 (3686), Cdc37 (10218-1-AP), cRaf (2330), Phos-cRaf (2330), BRaf (2330), Phos-bRaf (2330), c-Myc (5605), p90RSK (9326), Phos-p90RSK (9326), Hsp90 (4877), EGFR (2232), Phos-EGFR (4407), Akt1 (4691), phos-Akt308 (4056), and phos-Akt473 (4060) were purchased from Cell Signaling Technology (Danvers, MA, USA).

### 4.2. Cell Culture and Cell Viability Assay

The malignant melanoma cell lines, including SK-MEL-28, A875, B16, A375, and A2058 were obtained from the Chinese Center for Type Culture Collection (Wuhan, China) and cultured in DMEM (Dulbecco’s modified Eagle’s medium) with 10% FBS (fetal bovine serum) and streptomycin. The cytotoxicity assay was performed with the MTT method, as previously described [69,70,71].

### 4.3. Western Blotting Analysis

For western blotting, the total protein extracts from each sample were processed as previously described [72,73,74,75]. The total proteins were loaded into the SDS-PAGE (sodium dodecyl sulfate-polyacrylamide) gel and separated by their molecular weight via electrophoresis (PAGE). The separated proteins were then transferred to a PVDF membrane and then incubated by corresponding membrane react pathways (Millipore, Burlington, MA, USA), and the transferred membranes reacted with primary antibodies and HRP (horseradish peroxidase)-conjugated secondary antibodies (1:10,000). An enhanced chemiluminescence (ECL) WB substrate (Millipore, MA, USA) was used for identification of the expression profiles of target proteins.

### 4.4. Immunohistochemistry and Immunofluorescent Assays

The tumor tissue slices were immersed into EDTA antigenic retrieval buffer (pH 8.0) or citrate buffer (pH 6.0), and the antigen was recovered by microwave. The slides were then incubated with the corresponding primary antibody for 30–40 min at 37 °C. Normal anti-rabbit or anti-mouse IgG was used as a negative control group. The slide was treated with an HRP polymer coupled with the second antibody for 30 min, then developed with diaminobenzidine solution for immunohistochemistry analysis. For immunofluorescent (IF) assays, the slides were treated by fluorescein combined with a secondary antibody, and detected under a fluorescence microscope.

### 4.5. Protein co-IP Assay

The cells were cleaved in each group, followed by centrifugation of the supernatant, and addition of Hsp90 or EGFR primary antibody. Subsequently, the target proteins and their client proteins were collected using immunomagnetic beads. After centrifugal and cleaning, the amount of protein extracted from each group was determined by western blotting to demonstrate further whether there was an interaction between Hsp90 and EGFR.

### 4.6. Animal Models

The anti-tumor activity of DHP1808 in vivo was studied as per prescribed care guidelines. All animal experimental studies were approved by the Animal Ethics Committee of Animal Experimentation at Chengdu University of Traditional Chinese Medicine (project identification code: 2019-06). The 6–8 week-old SPF (specific pathogen-free) nude mice were purchased from Beijing Huafukang Biotechnology Co., Ltd. The preliminary safety evaluation of the anti-tumor activity of ENT-4G in vivo was carried out on A375 subcutaneous xenograft models. A375 cells were injected subcutaneously into the dorsal side of mice as a single cell suspension (5 × 10^5^/100 μL) in PBS. After the injection, tumor size was measured every two days. Tumor volume was calculated according to *Vol* = *a* × *b*^2^ × 0.5 (*a*, *b* represent the large and small diameter tumor tissue, respectively). The tumor tissue was stripped, fixed with formalin, embedded in paraffin, and sliced; the sections were then stained with TUNEL, EGFR, Akt, pAkt473, and Ki67 for further histological examination.

## Figures and Tables

**Figure 1 ijms-21-01845-f001:**
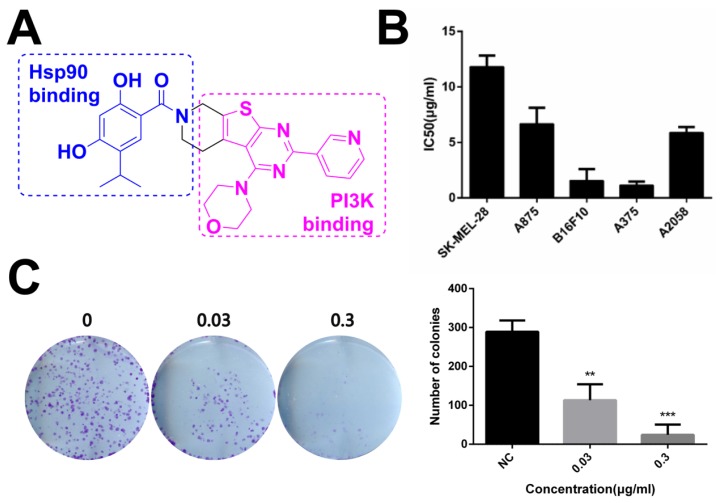
DHP1808 suppresses skin melanoma cell proliferation and clonogenic potential. (**A**) Chemical structure of dual inhibitor DHP1808; (**B**) Cells were incubated with various concentrations (1.25, 2.5, 5, 10, 20 or 40 μg/mL) of DHP1808 for 24 h. Mean IC_50_ values of DHP1808 against five melanoma cell lines after 24 h incubation; (**C**) Cells were incubated with various concentrations (0.03 or 0.3 μg/mL) of DHP1808 for 5 days. DHP1808 potently inhibited colony formation. ** *p* < 0.01, *** *p* < 0.001 versus the control group.

**Figure 2 ijms-21-01845-f002:**
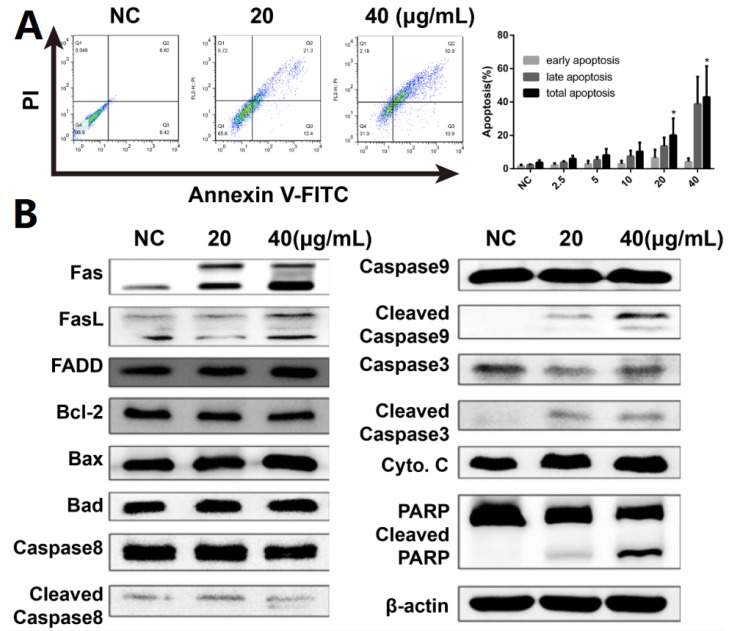
(**A**). A375 cells were incubated with various concentrations (0, 20, or 40 μg/mL) of DHP1808 for 24 h. Cell death were examined by Annexin V/PI double stained assay; (**B**). A375 cells were incubated with various concentrations (0, 20, or 40 μg/mL) of DHP1808 for 24 h. The expression levels of apoptosis-related proteins were determined by western blot analysis. Data represent means ± SD at least three independent experiments, * *p* < 0.05 versus the control group.

**Figure 3 ijms-21-01845-f003:**
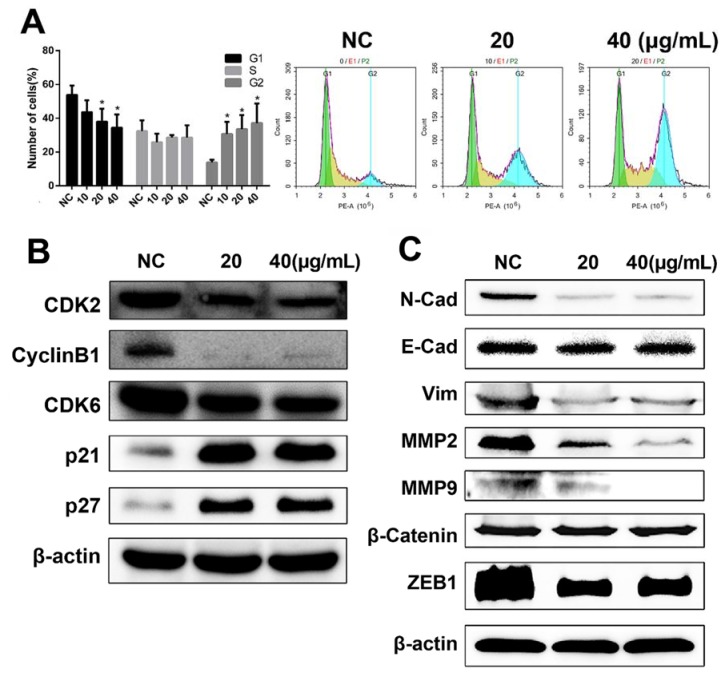
(**A**). A375 cells were incubated with various concentrations (0, 20 or 40 μg/mL) of DHP1808 for 24 h; the percentages on different phases of the cell cycle, G1: green, G2: blue, S: yellow. (**B**) A375 cells were incubated with various concentrations (0, 20 or 40 μg/mL). The expression levels of cell cycle related proteins were determined by western blot analysis. (**C**) A375 cells were incubated with various concentrations (0, 20, or 40 μg/mL). The levels of EMT-related proteins were determined by western blot analysis. Data represent means ± SD at least three independent experiments, * *p* < 0.05 versus the control group.

**Figure 4 ijms-21-01845-f004:**
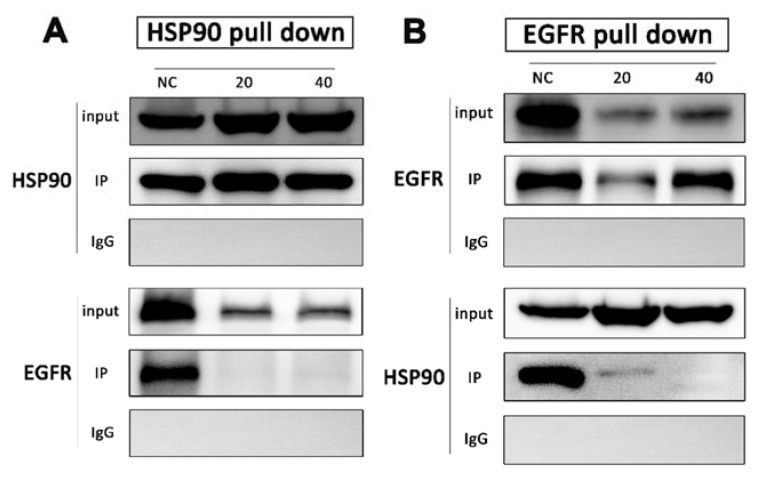
A375 cells were treated with DHP1808 for 24 h before lysis. (**A**,**B**) HSP90 and EGFR were immunoprecipitated from the lysates, respectively, and samples were analyzed by immunoblotting with anti-HSP90 and anti-EGFR antibodies.

**Figure 5 ijms-21-01845-f005:**
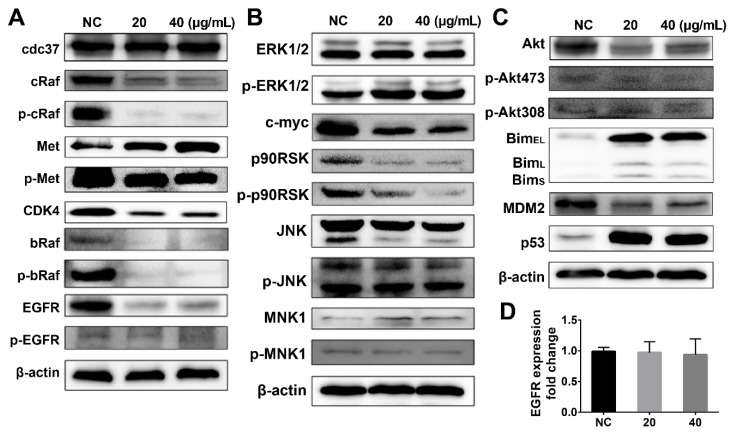
(**A**) Cells were incubated with various concentrations (0, 20, or 40 μg/mL) of DHP1808 for 24 h. Expression or phosphorylation levels of HSP90 client proteins were determined by western blot analysis; (**B**) Cells were incubated with various concentrations (0, 20, or 40 μg/mL) of DHP1808 for 24 h. Expression or phosphorylation levels of proteins associated with MAPK signaling pathway were determined by western blot analysis; (**C**) Cells were incubated with various concentrations (0, 20, or 40 μg/mL) of DHP1808 for 24 h. Expression or phosphorylation level of proteins associated with the PI3K-Akt signaling pathway was determined by western blot analysis; (**D**). The fold change of EGFR expression was determined by qPCR analysis.

**Figure 6 ijms-21-01845-f006:**
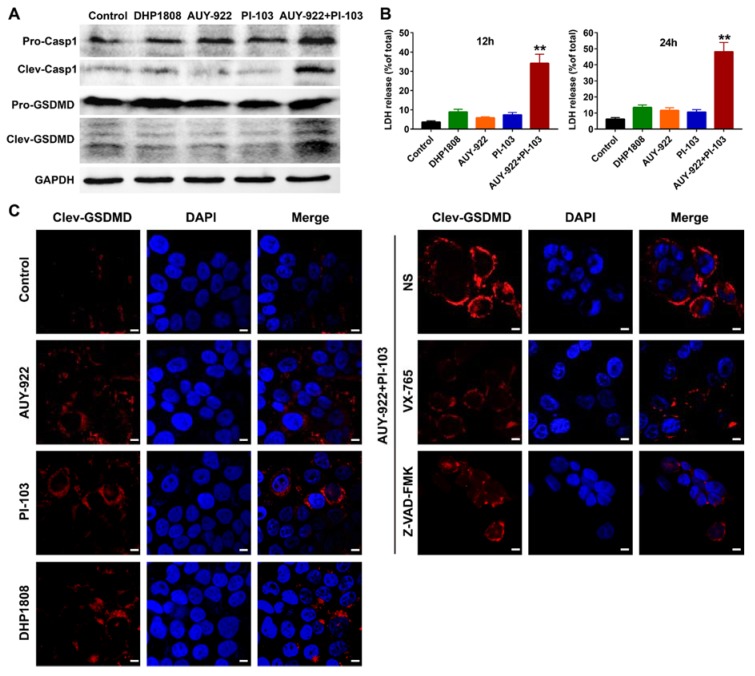
DHP1808 induced less pyroptosis than the combination of HSP90 and PI3K inhibitors. (**A**) WB analysis of the whole protein and cleaved forms of caspase-1 and GSDMD in A375 melanoma cells. (**B**) The release of LDH in A375 cells after 12 h or 24 h of drugs incubation, ** *p* < 0.01 versus the control group. (**C**) The immunofluorescent assay of cleaved GSDMD in A375 cells. The concentration of DHP1808, AUY-922, and PI-103 were set to 20 μM, scale bar: 6 μm.

**Figure 7 ijms-21-01845-f007:**
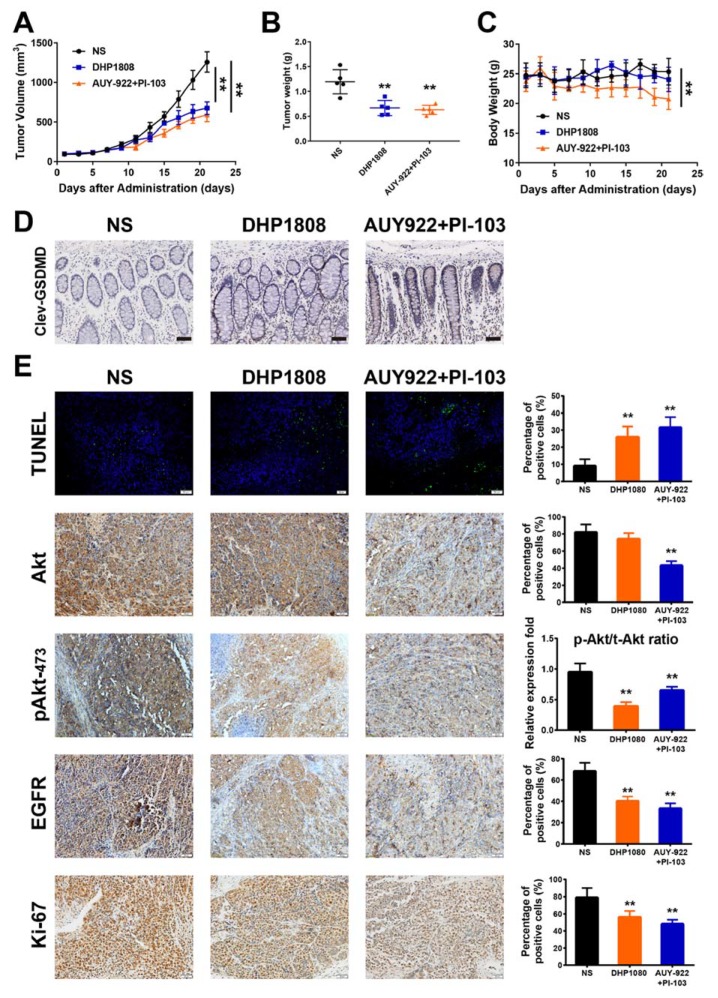
Inhibition effects of DHP1808 in a xenograft model. (**A**) Mice bearing A375 melanoma xenograft were treated with NS control, 30 mg/Kg DHP1808, and 30 mg/Kg AUY-922 plus 10 mg/Kg PI-103 once a day. Tumor volume data were presented as mean ± SD. (**B**) Tumor weight change of mice in different group. (**C**) Changes on mice bodyweight in each group. (**D**) IHC analysis of cleaved GSDMD in intestine tissues, scale bar: 100 μm. (**E**) TUNEL staining and IHC analysis of Akt, pAkt473, EGFR and Ki-67. Representative images of positive staining are shown, scale bar: 50 μm. ** *p* < 0.01 compared to control group. The intraperitoneal administration of 30 mg/Kg DHP1808, and 30 mg/Kg AUY-922 plus 10 mg/Kg PI-103 were performed.

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
