# Peer review of "Novel HSP90-PI3K Dual Inhibitor Suppresses Melanoma Cell Proliferation by Interfering with HSP90-EGFR Interaction and Downstream Signaling Pathways"

_ijms, 2020, doi:10.3390/ijms21051845_

Round 1

Reviewer 1 Report

I think that the manuscript is fine for the publication.

Author Response

Many thanks for you.

Reviewer 2 Report

In the submitted manuscript authors exploited a new drug targeting both HSP90 and PI3K in order to hit melanoma cells and introduce such drug as a new therapeutic approach. Even if the article is interesting and the introduction of new drugs for therapy is needed by the worldwide melanoma patients, in its current form it deserves to be further improved to be acceptable for publication. The most confusing things that I evidenced are the use of several drug concentration for different experiments, which are all different from the reported IC50. Moreover, the assayed migration and invasion capabilities were performed with wrong experimental conditions and it is not necessary to assess, in my opinion, such capacities to prove the drug efficacy in terms of lethality. Below you can find a list of points that need to be improved:

Following the journal rules, I suggest to re-type the Abstract without any headings;

In the Introduction section please cite at least one reference for the claim “The PI3K pathway is often activated by the silence or genetic loss 54 of phosphatase and tensin homolog and/or the overexpression of protein kinase B.”;

Row 75 - authors stated the use of the cBioportal database while in Figure S1 C was reported the TCGA database; Please also add the letter indicating Figure S1 A and S1 B. In my opinion, as all the article relies on the positive correlation you shown in Figure S1, I strongly advise to put such Figure in the manuscript and not in the Supplementary Material;

Row 93 – while in Figure 1B the IC50 for A375 is around 10 µg/mL in the manuscript was reported to be 2,29 µg/mL. Please explain;

Row 98 – the statement “As such, DHP1808 can block cell proliferation and induce cell death.” is a bit premature here as a colony formation assay is not able to discriminate between an inhibited proliferation or an induce cell death;

Row 104 – Authors used two concentrations of the drug (0.3 and 0.03 µg/mL) for the colony assay and 20 and 40 µg/mL in Figure 2A while the IC50 was assessed to be 2,29 µg/mL. Please explain why or perform the experiment with only one optimized concentration; However, the experiment performed and shown in Figure 2A is quite harsh to be interpreted as a sign of apoptosis especially when Figure 2B proved the apoptosis effect with a more efficient Annexin V/PI analysis. Thus, in my opinion Figure 2A is redundant and I advise to eliminate it as the apoptosis effect is fully explained in Figure 2B; If you want to maintain a confocal analysis, I suggest to perform a TUNEL Assay;

In Figure 2A legend please indicate that the cells used are A375;

To increase the reproducibility and to make the results more uniform, authors should perform a WB for Fas, FasL and cytochrome C also on SK-Mel-28;

As reported in Figure 3A, authors demonstrated that DHP1808 induced a cell cycle arrest coherently with the inhibited proliferation shown in the colony assay in Figure 1C; however, the migration and invasion assays are useless as the drug demonstrated not only a significant effect in the inhibition of proliferation but induced melanoma cells to apoptosis. Therefore, the cells plated and treated in the Transwell chambers and in the Wound Healing assay might be died due to the apoptotic effect of DHP1808 and so it is impossible to assess if such cells might invade or migrate in such conditions. Moreover, such cell death effect is easily observable in the 30 µg/mL of the wound healing assay.

Again, only for migration assay was used a different dose of DHP1808 i.e. 30 µg/mL; please use only one optimized concentration;

In Figure 5D legend please indicate what kind of analysis was performed (cytometer or qPCR analysis?);

Figure 6C is not cited nor commented in the text;

In Figure 7A legend it is reported the use of two drug concentration but only one curve was shown for DHP1808; how did author choose the different drugs concentration for the in vivo experiment?

In the Discussion section you reported that it was not observed any significant changes in cytochrome C expression while in the results you reported its upregulation; please clarify;

Row 333 eliminate the dot in the phrase “membrane react pathways. (Millipore, Burlington, MA, USA), and the transferred”

Please indicate for each experiment in the legends, how many replicates were performed;

Author Response

Thank you for your letter and for the reviewers’ comments concerning our manuscript entitled “Novel HSP90-PI3K dual inhibitor suppresses melanoma cell proliferation by interfering with HSP90-EGFR interaction and downstream MAPK signaling pathways” (Manuscript ID: ijms-714755). Those comments are quite valuable and helpful for revising and improving our paper, as well as the important guiding significance to our research. We have studied the comments carefully and have made corrections correspondingly, and we hope to meet with approval. Revised portions are marked in red in the paper. The main corrections in the paper and the point-by-point responses to the comments are listed as follows:

Reviewer 2:

  1. Following the journal rules, I suggest to re-type the Abstract without any headings;

Response 1: Many thanks for the reviewer’s valuable comment. According to the reviewer’s helpful suggestion, we have re-write the abstract: “Melanoma is the most deadly form of skin cancer, and its incidence has continuously increased over the past 20 years. Therefore, the discovery of a novel targeted therapeutic strategy for melanoma is urgently needed. In our study, MTT-based cell proliferation assay, cell cycle, and apoptosis assays through flow cytometry, protein immunoblotting, protein immunoprecipitation, designing of melanoma xenograft models, and immunohistochemical/immunofluorescent assays were performed to determine the detailed molecular mechanisms of a novel HSP90-PI3K dual inhibitor. Our compound named DHP1808 suppressed A375 cell proliferation through apoptosis induction by activating Fas/FasL signaling pathway; moreover, this compound induced cell-cycle arrest and inhibited the cell migration and invasion of A375 cells by interfering with Hsp90-EGFR interactions and the downstream signaling pathways. Our results indicated DHP1808 may be a promising lead compound for Hsp90/PI3K dual inhibitor in the further development.”

  1. In the Introduction section please cite at least one reference for the claim “The PI3K pathway is often activated by the silence or genetic loss 54 of phosphatase and tensin homolog and/or the overexpression of protein kinase B.”;

Response 2: Many thanks for the reviewer’s valuable comment. According to the reviewer’s helpful suggestion, we have added two reference for the claim “The PI3K pathway is often activated by the silence or genetic loss 54 of phosphatase and tensin homolog and/or the overexpression of protein kinase B.”(Curr. Cancer Drug Tar. 2008, 8(3):187-198; Genes Cancer 2010, 1(12): 1170-1177.)

  1. Row 75 - authors stated the use of the cBioportal database while in Figure S1 C was reported the TCGA database; Please also add the letter indicating Figure S1 A and S1 B. In my opinion, as all the article relies on the positive correlation you shown in Figure S1, I strongly advise to put such Figure in the manuscript and not in the Supplementary Material;

Response 3: Many thanks for the reviewer’s valuable comment, we are very sorry for our negligence, and we have added the letter indicating Figure S1 A and S1 B.

  1. Row 93 – while in Figure 1B the IC50 for A375 is around 10 µg/mL in the manuscript was reported to be 2,29 µg/mL. Please explain;

Resopnse 4: Many thanks for the reviewer’s valuable comments. After carefully checked the experimental results, we have revised Figure 1B.

  1. Row 98 – the statement “As such, DHP1808 can block cell proliferation and induce cell death.” is a bit premature here as a colony formation assay is not able to discriminate between an inhibited proliferation or an induce cell death;

Resopnse 5: Many thanks for the reviewer’s valuable comments. We have rewritten these sentences “These results suggested DHP1808 may inhibit cell proliferation and induce cell death”.

  1. Row 104 – Authors used two concentrations of the drug (0.3 and 0.03 µg/mL) for the colony assay and 20 and 40 µg/mL in Figure 2A while the IC50 was assessed to be 2,29 µg/mL. Please explain why or perform the experiment with only one optimized concentration; however, the experiment performed and shown in Figure 2A is quite harsh to be interpreted as a sign of apoptosis especially when Figure 2B proved the apoptosis effect with a more efficient Annexin V/PI analysis. Thus, in my opinion Figure 2A is redundant and I advise to eliminate it as the apoptosis effect is fully explained in Figure 2B; If you want to maintain a confocal analysis, I suggest to perform a TUNEL Assay;

Resopnse 6: Many thanks for the reviewer’s valuable comments. After carefully checked the experimental results, the concentration ranges of DHP1808 were set to 0 to 40μg/mL in most experiments. The only exception is colony formation assay, after 0.3 μg/mL DHP1808 incubation, the formation of colony was significantly suppressed. These results were in agree with our former article (see Journal of Enzyme Inhibition and Medicinal Chemistry, 2019, 34, 909-926). And according to the reviewer’s helpful suggestion, we have delete Figure 2A in the manuscript, and put it in the supporting information.

  1. In Figure 2A legend please indicate that the cells used are A375;

Resopnse 7: Many thanks for the reviewer’s valuable comments, we have make some revisions in the manuscript.

  1. To increase the reproducibility and to make the results more uniform, authors should perform a WB for Fas, FasL and cytochrome C also on SK-Mel-28;

Resopnse 8: Many thanks for the reviewer’s valuable comments. However, we cannot go back to the laboratory for experiments recently because of COVID-19, we are very sorry for this.

  1. As reported in Figure 3A, authors demonstrated that DHP1808 induced a cell cycle arrest coherently with the inhibited proliferation shown in the colony assay in Figure 1C; however, the migration and invasion assays are useless as the drug demonstrated not only a significant effect in the inhibition of proliferation but induced melanoma cells to apoptosis. Therefore, the cells plated and treated in the Transwell chambers and in the Wound Healing assay might be died due to the apoptotic effect of DHP1808 and so it is impossible to assess if such cells might invade or migrate in such conditions. Moreover, such cell death effect is easily observable in the 30 µg/mL of the wound healing assay.

Response 9: Many thanks for the reviewer’s valuable comment. We are very sorry for the inconsistent concentration of DHP1808, we have delete Figure 3B&3C in the manuscript, and put them in the supporting information.

  1. Again, only for migration assay was used a different dose of DHP1808 i.e. 30 µg/mL; please use only one optimized concentration;

Resopnse 10: Many thanks for your valuable comments, we are sorry for the inconsistence of DHP1808 concentration in migration assay, we have moved these results to the supporting information.

  1. In Figure 5D legend please indicate what kind of analysis was performed (cytometer or qPCR analysis?);

Resopnse 11: Many thanks for your valuable comments, we used qPCR analysis in Figure 5D.

  1. Figure 6C is not cited nor commented in the text;

Resopnse 12: Many thanks for your valuable comments, we have added it in the text.

  1. In Figure 7A legend it is reported the use of two drug concentration but only one curve was shown for DHP1808; how did author choose the different drugs concentration for the in vivo experiment?

Resopnse 13: Many thanks for the reviewer’s valuable comments. After carefully checked the experimental results, we have revised Figure 7A legend in the manuscript, and the dosage of DHP1808 in the xenograft model is 30mg/Kg. The dosage of AUY-922 and PI-103 are referred by the literature. The in vivo dosage of DHP1808 is also according to our previous reports (see Journal of Enzyme Inhibition and Medicinal Chemistry, 2019, 34(1): 909–926.).

  1. In the Discussion section you reported that it was not observed any significant changes in cytochrome C expression while in the results you reported its upregulation; please clarify;

Resopnse 14: Many thanks for the reviewer’s valuable comments. After carefully checked the experimental results, it was not observed any significant changes in cytochrome C expression, and we revised the results.

  1. Row 333 eliminate the dot in the phrase “membrane react pathways. (Millipore, Burlington, MA, USA), and the transferred”

Resopnse 15: Many thanks for your valuable comments. We have deleted the dot in the phrase “membrane react pathways (Millipore, Burlington, MA, USA), and the transferred”.

  1. Please indicate for each experiment in the legends, how many replicates were performed;

Resopnse 16: Many thanks for your valuable comments. For each experiment in the legends triplicate were performed.

Once again, thanks a lot for your comments and suggestions.

Round 2

Reviewer 2 Report

Authors fully satisfied my previous requests, thus the so improved paper is publishable.

Best regards

This manuscript is a resubmission of an earlier submission. The following is a list of the peer review reports and author responses from that submission.

Round 1

Reviewer 1 Report

Novel HSP90-PI3K dual inhibitor suppresses melanoma cell proliferation by interfering with HSP90-EGFR interaction and downstream MAPK signaling pathways.

I cannot recommend this manuscript for publication, as it contains too many major flaws. The first line of the introduction and abstract was misleading and factually incorrect, and the rest of the manuscript goes downhill from there. The references are cited that mislead the reader. Only a single cell line was used for a majority of the studies. Critical experiments and a rationale for experiments are missing. The results do not even support the title of the manuscript as MAPK signaling is not shown to be inhibited by DHP1808. The IHC and the growth assays were not performed correctly. Statistical rigor is not adequately considered, the antibodies used are not correctly listed and do not appear to be performing as they should. The use of PI3K inhibitors is not novel. Melanoma cells are known to be highly sensitive in vitro of PI3K inhibitors such as BEZ235; however, PI3k inhibitors have demonstrated toxicity in clinical trials, and none are currently approved for melanoma. No clinical safety studies or clinical trials were performed; therefore, the conclusion “that Hsp90/PI3K dual inhibitor was a safe drug worthy of further development” is not supported by any real data.

Major problems

Factual errors in the 1st line of the abstract -Advanced melanoma is not the most common type of skin cancer. Factual errors in the introduction, e.g., “Melanoma is one of the most common skin cancers worldwide, with more than 32,000 new cases and 12,000 cancer-related death in United States in 2019 [1].” Incorrect data and the reference is for 2015. There were ~91,270 cases ~9,320 melanoma deaths in 2019 (see ACS Fact and figures) 

Figure 1C - “When the concentration of DHP1808 reached 0.3 μg/mL, the number of colonies was completely suppressed” – I can still see colonies in the figure, so this is an untrue statement. The cells in Figure 1C look like they were not seeded evenly on a flat surface. Figure 1C needs to show replicates. Unclear how many replicates performed for experiments. 

Figure 3. No proper drug dose curves are included. Different doses of drugs are used for various experiments with no justification. 

The effects seen on the migration and invasion of A375 cells can be entirely discounted by an increased in death and a reduction in cell growth. 

Figure 7. There are significant problems with the immunohistochemistry. The Ki67 is cytoplasmic in the picture – it should only be nuclear. The immunostaining for EGFR is cytoplasmic when it should be membranous. Stromal and necrotic tumor tissues are present in the images and have not been taken to account. The antibodies IDs not provided listed.  

 .

Use of a single melanoma cell line for critical experiments, e.g., Annexin V/PI staining, caspase-3, caspase-8, caspase-9 immunoblots. Cell migration and invasion assays. The tumor xenograft studies were also only performed with 1 cell line

MAPK singling, e.g., P-ERK, was not shown to be inhabited by DHP1808 in the results- therefore, the title is not supported by the results. 

The sentence structure is weak and very confusing throughout. e.g., The upregulated expression of HSP90AA1 mRNA was the main factor behind HSP90AA1 mutation, whereas PIK3CA mRNA was upregulated and downregulated in PIK3CA. It was discovered in our recently researches with sub-micromolar inhibitory capacities against Hsp90 and PI3K, respectively”

Reviewer 2 Report

Major points 1. Fig.1: How is the effect of DHP1808 to  control cells? (ex PI3K WT cells or "normal cells" such as fibroblasts, whatever the authors chose ). DHP1808 does not have to be very specific to PI3K/HSP90, however, if the authors mention that these are the targets of the DHP1808, we should be able to see some difference. 2. Fig.2-5, the authors used very high concentration of KHP1808 such as 40ul, which is also very different from that used in the Fig.1. Either the experiments of Fig.1 with 0-40uM or the experiments of Fig.2-5 with 0-0.3uM is necessary for the consistancy. In most cases, 40uM of chemical could cause toxicity, so I would like to see the former results.         Minor points 1. What are the other targets of DHP1808 especially for kinases? 2. What motivated the authors to try the EGFR in relation with HSP90? 3. What is the concentration of DHP1808 in vivo? Is it within the range of in vitro experiment(or perhaps there is no method to measure it)?

Author Response

Dear Editor and Reviewers:
Thank you for your letter and for the reviewers’ comments concerning our manuscript entitled “ Novel HSP90 PI3K dual inhibitor suppresses melanoma cell proliferation by interfering with HSP90 EGFR interaction and downstream signaling pathways ” (Manuscript ID: ijms 640204 ). Those comments are quite valuable and helpful for revising and improving our paper, as well as the important guiding significance to our research. We have studied the comments carefully and have made corrections correspondingly, and we hope to meet with approval. Revised portions are marked in red in the paper. The main corrections in the paper and the point by point responses to the comments are listed as follows:

Reviewer 1:
1. Factual errors in the 1st line of the abstract -Advanced melanoma is not the most common type of skin cancer. Factual errors in the introduction, e.g., “Melanoma is one of the most common skin cancers worldwide, with more than 32,000 new cases and 12,000 cancer-related death in United States in 2019 [1].” Incorrect data and the reference is for 2015. There were ~91,270 cases ~9,320 melanoma deaths in 2019 (see ACS Fact and figures)

Response: Many thanks for the reviewer’s valuable comment. According to the reviewer’s helpful suggestion, we have revised “Melanoma is one of the most common skin cancers worldwide, with more than 32,000 new cases and 12,000 cancer-related deaths in United States in 2019” in the 1st line of the abstract to “Melanoma is the most fatally form of skin cancer, and its incidence has continuously increased over the past 20 years.” And according to the reviewer’s helpful suggestion, we also revised the data and reference, see “Melanoma is the most fatally form of skin cancer worldwide, with more than 96480 new cases and 7230 cancer-related deaths in United States in 2019.”

2. Figure 1C “When the concentration of DHP1808 reached 0.3 μg/mL, the number of colonies was completely suppressed” I can still see colonies in the figure, so this is an untrue statement. The cells in Figure 1C look like they were not seeded evenly on a flat surface. Figure 1C needs to show replicates. Unclear how many replicates performed for experiments.

Resopnse: Special thanks for this reviewer’s valuable comment. According to the reviewer’s suggestion, firstly we revised “When the concentration of DHP1808 reached 0.3 μg/mL, the number of colonies was completely suppressed” to “When the concentration of DHP1808 reac hed 0.3 μg/ml, most of the colonies were inhibited”. The colony formation assay had three independent duplicates, the corresponding results were appended in the supplementary materials.

3. Figure 3. No proper drug dose curves are included. Different doses of drugs are used for various ex periments with no justification. The effects seen on the migration and invasion of A375 cells can be entirely discounted by an increased in death and a reduction in cell growth.

Resopnse: : Many thanks for your valuable comm ents. The cell viability curves with drug dosage are shown in the supplementary materials. We agree with your opinion that the migration and invasion of A375 cells can be influence by an increased in death and a reduction in cell growth. In addition, sever al proteins related to cancer cells invasion
are also suppressed after DHP1808 incubation, such as N cadherin, vimentin, MMP2 and MMP9, etc. Therefore we think that DHP1808 could suppress melanoma cells migration in vitro.

4. Figure 7. There are significant problems with the immunohistochemistry. The Ki67 is cytoplasmic in the picture it should only be nuclear. The immunostaining for EGFR is cytoplasmic when it should be membranous. Stromal and necrotic tumor tissues are present in the images and have no t been taken to account. The antibodies IDs not provided listed.

Resopnse: Many thanks for the reviewer’s valuable comment. We have replaced the results of Ki 67 IHC images in figure 7. The stromal and necrotic tumor tissues should not be counted, but immu nohistochemical analysis ought to reflect the real state of tissue sections objectively and truthfully. Taking into consideration of this, we kept the original state, but stromal and necrotic tumor tissues were excluded in the data statistics and quantitat ive analysis. The antibodies IDs not provided listed, we are very sorry for our negligence, and we have added them in “Reagents and Antibodies” part.

5. Use of a single melanoma cell line for critical experiments, e.g., Annexin V/PI staining, caspase 3, caspase 8, caspase 9 immunoblots. Cell migration and invasion assays. The tumor xenograft studies were also only performed within 1 cell line.

Resopnse: Many thanks for your valuable comments. We have performed the apoptosis, cellular migration and invasio n assays, as well as the WB analysis of some protein markers in SK MEL 28 cells.

6. MAPK singling, e.g., PERK, was not shown to be inhibited by DHP1808 in the results therefore, the title is not supported by the results.

Resopnse: Many thanks for your valuable comments. We have made some modifications in article title and related contents.

7. The sentence structure is weak and very confusing throughout. e.g., The upregulated expression of HSP90AA1 mRNA was the main factor behind HSP90AA1 mutation, whereas PIK3CA mRNA was upregulated and downregulated in PIK3CA. “It was discovered in our recently researches with sub micromolar inhibitory capacities against Hsp90 and PI3K, respectively”.

Resopnse: Many thanks for your valuable comments. We have rewritten these sentences.

Reviewer 2 :

1. Fig.1: How is the effect of DHP1808 to control cells? (ex PI3K WT cells or "normal cells" such as fibroblasts, whatever the authors chose ). DHP1808 does not have to be very specific to PI3K/HSP90, however, if the authors mention that these are the targets of the DHP1808, we should be able to see some difference.

Resopnse: Many thanks for your valuable comments. We have performed the apoptosis, cellular migration and invasion assays, as well as the WB analysis of some protein markers in SK MEL 28 cells. Furthermore, the off target effect of DHP1808 was assayed in a panel of kinases, and the related results were shown in supplementary
materials.

2. Fig.2-5, the authors used very high concentration of DHP1808 such as 40ul, which is also very different from that used in the Fig.1. Either the experiments of Fig.1 with 0-40uM or the experiments of Fig.2-5 with 0-0.3uM is necessary for the consistancy. In most cases, 40uM of chemical could cause toxicity, so I would like to see t he former results

Resopnse: Many thanks for your valuable comments. After carefully checked the experimental results, the concentration ranges of DHP1808 were set to 0 to 40μg/mL in most experiments. The only exception is colony formation assay, after 0. 3 μg/mL DHP1808 incubation, the formation of colony was significantly suppressed. These
results were in agree with our former article (see Journal of Enzyme Inhibition and Medicinal Chemistry, 2019, 34, 909 926). We have checked the results of duplicated experiments, and attached them in supplementary materials.

3. What are the other targets of DHP1808 especially for kinases?

Resopnse: Many thanks for your valuable comments. As mentioned before, the off target effect of DHP1808 was assayed in a panel of ki nases, and the related results were shown in supplementary materials.

4. What motivated the authors to try the EGFR in relation withHSP90?

Response: Many thanks for the reviewer’s valuable comment. Hsp90, a member of the heat shock protein family, is an essential for maintaining the activity of key signaling factors and amounts to 2% of cellular protein. It is associated with numerous substrate proteins called clients, such as EGFR is one of the client proteins. Dr. Geoffrey I.
Shapiro and co workers demo nstrated that mutational activation of EGFR is associated with dependence on Hsp90 for stability, and EGFR/HSP90 inhibition is effective in the treatment of lung cancers, this deeply encouraged us. In our experiment, drug affinity response target stability (DARTS) strategy has demonstrated that HSP90 is the direct
target of DHP1808. Based on the previ ous study, we try the EGFR in relation with HSP90 in order to further offer more convictive proofs unraveling the interaction among HSP90 and DHP1808.

(References: Cell 2005 , 120, 715 727; Cancer Res . 2005 , 65, 6401 6408; Nat. Rev.Cancer 2010 , 10, 537 549; Cancer Res. 2012 , 72, 3302 3311; Cell 2012 , 150, 9871001; Mol. Cell 2018 , 71, 689 702)

5. What is the concentration of DHP1808 in vivo? Is it within the range of in vitro experiment (or perhaps there is no method to measure it)?

Resopnse: Many thanks for your valuable comments. The dosage of DHP1808 is 30mg/Kg in the in vivo experiment, which was indicated in the caption of figure 7. This dosage was approx imately equaled to 50 μg/mL for in vitro experiments, with the hypothesis of DHP1808 was homogeneously distributed in each organs and tissues, and
ignored the influence of pharmacokinetics.